

# *Methylobacterium*, a major component of the culturable bacterial endophyte community of wild *Brassica* seed

Davood Roodi[1,2,3], James P. Millner[1], Craig McGill[1], Richard D. Johnson[3], Ruy Jauregui[4] and Stuart D. Card[3]

[1] School of Agriculture & Environment, Massey University, Palmerston North, Manawatu, New Zealand
[2] Agricultural Research, Education and Extension Organization (AREEO), Seed and Plant Improvement Institute, Karaj, Alborz, Iran
[3] Forage Science, AgResearch Limited, Grasslands Research Centre, Palmerston North, Manawatu, New Zealand
[4] Knowledge & Analytics, AgResearch Limited, Grasslands Research Centre, Palmerston North, Manawatu, New Zealand

Corresponding author
Stuart D. Card,
Stuart.Card@agresearch.co.nz

## ABSTRACT

**Background:** Plants are commonly colonized by a wide diversity of microbial species and the relationships created can range from mutualistic through to parasitic. Microorganisms that typically form symptomless associations with internal plant tissues are termed endophytes. Endophytes associate with most plant species found in natural and managed ecosystems. They are extremely important plant partners that provide improved stress tolerance to the host compared with plants that lack this symbiosis. Plant domestication has reduced endophyte diversity and therefore the wild relatives of many crop species remain untapped reservoirs of beneficial microbes. *Brassica* species display immense diversity and consequently provide the greatest assortment of products used by humans from a single plant genus important for agriculture, horticulture, bioremediation, medicine, soil conditioners, composting crops, and in the production of edible and industrial oils. Many endophytes are horizontally transmitted, but some can colonize the plant's reproductive tissues, and this gives these symbionts an efficient mechanism of propagation *via* plant seed (termed vertical transmission).

**Methods:** This study surveyed 83 wild and landrace *Brassica* accessions composed of 14 different species with a worldwide distribution for seed-originating bacterial endophytes. Seed was stringently disinfected, sown within sterile tissue culture pots within a sterile environment and incubated. After approximately 1-month, direct isolation techniques were used to recover bacterial endophytes from roots and shoots of symptomless plants. Bacteria were identified based on the PCR amplification of partial 16S rDNA gene sequences and annotated using the BLASTn program against the NCBI rRNA database. A diversity index was used as a quantitative measure to reflect how many different bacterial species there were in the seed-originating microbial community of the *Brassica* accessions sampled.

**Results:** Bacterial endophytes were recovered from the majority of the *Brassica* accessions screened. 16S rDNA gene sequencing identified 19 different bacterial species belonging to three phyla, namely Actinobacteria, Firmicutes and Proteobacteria with the most frequently isolated species being *Methylobacterium*

*fujisawaense, Stenotrophomonas rhizophila* and *Pseudomonas lactis*. *Methylobacterium* was the dominant genus composing 56% of the culturable isolated bacterial community and was common in 77% of accessions possessing culturable bacterial endophytes. Two selected isolates of *Methylobacterium* significantly promoted plant growth when inoculated into a cultivar of oilseed rape and inhibited the growth of the pathogen *Leptosphaeria maculans* in dual culture. This is the first report that investigates the seed-originating endophytic microorganisms of wild *Brassica* species and highlights the *Brassica* microbiome as a resource for plant growth promoting bacteria and biological control agents.

## INTRODUCTION

Endophytes are a diverse sub-group of microorganisms that reside inside the tissues of nearly every vascular plant and, for at least part of their life cycle, do not cause any immediate symptoms (*Card et al., 2016*; *Hardoim et al., 2015*; *Porras-Alfaro & Bayman, 2011*; *Wilson, 1995*). However, not all endophytes remain within their plant host throughout their entire life cycle. Additionally, some may change their behavior, from mutualistic to commensalism or even pathogenic, due to a change in the environment, during host senescence or when the host is stressed (*Aly, Debbab & Proksch, 2011*; *Fisher & Petrini, 1992*). Endophytes can be found in nearly every type of plant organ, in both vegetative (e.g., leaves, roots and shoots) and reproductive (e.g., flower and seed) tissues (*Rodriguez et al., 2009*). The presence of bacterial endophytes within the reproductive tissues has been reported for many plant species (*Mundt & Hinkle, 1976*), including coffee (*Vega et al., 2005*), cotton (*Adams & Kloepper, 1996*), cucumber (*Khalaf & Raizada, 2016*), eucalyptus (*Ferreira et al., 2008*), oilseed rape (*Granér et al., 2003*), maize (*Rijavec et al., 2007*), Norway spruce (*Cankar et al., 2005*), tobacco (*Mastretta et al., 2009*) and rice (*Elbeltagy et al., 2000*; *Okunishi et al., 2005*). These seed-originating bacterial endophytes may be disseminated from one generation to the next, persisting in the next population of plants (*López-López et al., 2010*) and is indicative of their ability to vertically transmit. Plant hosts harboring endophytes can gain additional advantageous traits, granting them an ecological advantage over individuals lacking these microorganisms and/or other plant species that occupy a similar ecological niche. These benefits include greater resistance to abiotic and biotic stresses (*Hallmann et al., 1997*; *Mastretta et al., 2006*) as well as plant growth promotion (*Azevedo et al., 2000*).

Modern *Brassica* cultivars were originally domesticated from species mostly originating from Europe (*Rakow, 2004*), although now many *Brassica* crops, particularly *Brassica napus* (oilseed rape), *Brassica rapa* (turnip) and *Brassica oleracea* (cabbage), are extensively cultivated throughout the world. These species are a major source of vegetables for human consumption and for forage, ornamental plants, condiments, medicinal crops, green manure, bioremediation, and as very important sources of edible and industrial oils

(*Dixon, 2007*; *Gómez-Campo, 1980*). A wide range of insect pests, such as aphids (*Brevicoryne brassica*), diamond back moth (*Plutella xylostella*) and flea beetles (*Phyllotreta* and *Psylliodes* spp.), in addition to several diseases, such as clubroot (caused by *Plasmodiophora brassicae*), phoma stem canker (caused by *Leptosphaeria maculans*) and sclerotinia stem rot (caused by *Sclerotinia sclerotiorum*) cause extensive damage to *Brassica* crops worldwide (*Kimber & McGregor, 1995*) with few or no control options available (*Granér et al., 2003*).

Most studies investigating endophytes of *Brassica* have focused on isolating microorganisms from the vegetative tissues of modern-day cultivars (*Germida et al., 1998*; *Narisawa, Tokumasu & Hashiba, 1998*; *Sheng et al., 2008*; *Sunkar & Nachiyar, 2013*; *Zhang et al., 2014*). However, this strategy may be restrictive as the diversity and frequency of endophytic species found in domesticated crops is assumed to be much lower than in their respective wild relatives (*Mousa et al., 2015*; *Putra, Rahayu & Hidayat, 2015*). Additionally, targeting endophytic species that are associated with the reproductive plant tissues (those microorganisms that are seed-borne or seed-transmitted) would greatly aid the marketing of potential commercial products (*Card et al., 2015*, *2016*). This study focused on developing a strategy for screening wild and landrace *Brassica* species for mutualistic, seed-originating endophytes that may offer beneficial traits to elite *Brassica* cultivars.

## MATERIALS AND METHODS

### Brassica germplasm

A total of 64 accessions (49 wild and 15 landraces) of *Brassica* (Table S1), encompassing a diverse number of species, with a worldwide distribution, were obtained from three international genebanks, namely The United States Department of Agriculture (USDA) via The Germplasm Resources Information Network (GRIN), The Leibniz-Institute of Plant Genetics and Crop Plant Research (IPK Gatersleben) and The Nordic Genetic Resource Centre (NordGen). These accessions were imported into the Margot Forde Germplasm Centre (MFGC), New Zealand's national genebank of grassland plants (on seed import permit no. 2015058982). A further 19 populations of wild *Brassica* were collected locally, within the Manawatu-Wanganui region situated in the lower half of the North Island of New Zealand. All 83 accessions were cataloged and stored at 0°C and 30% relative humidity within the MFGC.

### Screening for seed-originating bacterial endophytes

Two surface disinfection protocols were developed to remove non-target microorganisms (such as saprophytic microorganisms) associated with the seed surface of the aforementioned *Brassica* accessions. Initially all 83 accessions were surface disinfected using the following protocol: seeds were washed for five min in 5% aqueous Tween-20® solution (Sigma-Aldrich Inc., Auckland, New Zealand), two min in 70% ethanol, 10 min in 0.5% sodium hypochlorite, one min in 70% ethanol and rinsed three times in sterile water. To assess the efficacy of the surface disinfection protocols, $3 \times 20$ µL drops of water from the last rinse were plated onto nutrient agar (NA), (CM003, Oxoid Ltd., UK). Petri plates containing the NA were incubated for 2 weeks at 22 °C and inspected daily

for microbial growth with the aid of a dissecting microscope (Carl Zeiss (N.Z.) Ltd., New Zealand). For those accessions where saprophytic microorganisms were initially observed, a second surface disinfection protocol was applied whereby the same procedure listed above was repeated except for one modification; seeds were immersed for 10 min in 2% sodium hypochlorite rather than a 0.5% solution. Seed were then dried on filter paper (110 mm, Thermo Fisher Scientific Ltd., Auckland, New Zealand) within a sterile environment and stored at 4 °C in the dark. To induce germination, seeds were dipped into sterile 0.2% $KNO_3$ solution and immediately plated onto Petri plates containing 1.5% water agar (WA), with 10 seeds per plate. Petri plates were incubated at 4 °C in the dark for 72 h to break seed dormancy. Dormant accessions (as tested for 15 accessions) resulted in either zero or poor germination without this process and were subsequently transferred to a custom-built growth chamber at 22–25 °C and a 16/8 h (light/dark) photoperiod.

Seed and the subsequent seedlings were examined daily under a dissecting microscope and those exhibiting any obvious epiphytic microbial growth were discarded. After 2–3 days, 10 clean seedlings, from each accession, were transferred to sterile tissue culture pots, 98 mm diameter (2105646, Alto Ltd., New Zealand) containing Murashige & Skoog (MS) basal salts (*Murashige & Skoog, 1962*) with minimal organics (Sigma-Aldrich, New Zealand), plus 3% sucrose and 1.5% agar (*Ali et al., 2007*). Pots were placed in the growth chamber (with the same settings as described earlier) and visually assessed every day for a month using a dissecting microscope. Plants were discarded if they showed any disease symptoms or any saprophytic microbial growth. Four clean seedlings were finally selected from each accession and subsequently dissected into two components: shoot and root. These organs were further dissected into 2–3 $mm^2$ pieces using sterile forceps and a scalpel. Ten pieces per organ type from each seedling were transferred to Petri plates containing NA. Petri plates were incubated for 3 weeks at 22 °C in the dark and checked daily under a dissecting microscope for microbial growth. Bacterial colonies arising from dissected tissue pieces were selected, sub-cultured and checked for purity. Representative bacterial isolates were then sub-cultured onto fresh NA using a sterile loop and stored in 25% glycerol at −80 °C.

## Identification of seed-originating bacterial endophytes

Bacterial isolates were identified based on the PCR amplification of partial 16S rDNA gene sequences (*Weisburg et al., 1991*). PCR was performed directly on suspensions of each purified bacterial colony as follows: each colony was suspended in 10 μL Milli-Q® water in a standard 0.2 mL PCR tube (Axygen™, San Francisco, CA, USA) and frozen at −20 °C before being thawed and heated at 65 °C for 30 min. one μL of suspension was added to the PCR reaction containing five μL 10× PCR buffer, 1.5 μL $MgCl_2$ (50 mM), forward primer, 27F (5′-AGAGTTTGATCCTGGCTCAG,one μL, 10 μM), reverse primer R1497, (5′-CCTATATCGCCGGTAATT, one μL,10 μM), 0.4 μL dNTPS (25 mM), 0.25 μL Taq-polymerase and 39.85 μL sterile Milli-Q water to make a 50 μL PCR reaction. PCR was performed in a thermocycler (Bio-Rad C1000 Touch™, Bio-Rad Laboratories Inc., Hercules, CA, USA) with the following conditions: an initial step of 95 °C for 5 min was

followed by 36 cycles of 94 °C for 30 s, 56 °C for 60 s, 72 °C for 90 s and a final step of 72 °C for 10 min; PCR amplification products were confirmed by electrophoresis in a 1.5% agarose gel and purified using the DNA clean & concentrator kit (Zymo Research Corporation, Irvine, CA, USA) prior to Sanger sequencing (*Sanger & Coulson, 1975*) (New Zealand Genomics Ltd., New Zealand). The raw sequence ab1 files were imported into the software package Geneious Prime® version 2019.1.1 (Biomatters Ltd., Auckland, New Zealand) and were quality trimmed using an error probability of 0.05. Those sequences with a region of high quality greater or equal than 600 bp were kept and annotated using the BLASTn program against the NCBI rRNA database. The sequences were aligned using the multiple alignment program MAFFT (*Katoh & Standley, 2013*), and a Maximum Likelihood phylogenetic tree was generated using the software Mega X (*Kumar et al., 2018*) using a general time reversible model and validated by 100 bootstrap cycles. All sequence data were deposited in GenBank under file SUB6483552: MN629046–MN629135. Simpson`s diversity index (*Simpson, 1949*) was used as a quantitative measure to reflect how many different bacterial species there were in the seed-originating microbial community of the *Brassica* accessions sampled.

## Assessing plant growth promotion

Two isolates of *Methylobacterium*, namely *Methylobacterium fujisawaense* (isolate B82) and *Methylobacterium phyllosphaerae* (isolate B64), were selected due to their high tissue colonization rate in their original host plants. The bacteria were plated on NA and incubated for 2 weeks at 22 °C. For each isolate, cells were scraped from the Petri plates using a loop and transferred to an aqueous Tween-20® solution. Concentrations were adjusted to $10^9$ cells per mL using a hemocytometer. Oilseed rape, cv. King was selected as the novel host plant. Seeds were surface disinfected, as described earlier, and placed on a filter paper under a laminar flow cabinet to dry. They were then transferred to Petri plates containing 2% WA and incubated at 22 °C in a custom-made lighting room with 18/6 h (light/dark photoperiod) to initiate germination. The root tip of each seedling was excised with a sterile scalpel, dipped into the prepared bacterial suspension and transplanted into sterile plastic pots (7 × 15 cm) containing autoclaved potting mix (50% fine bark, 12.5% compost and 25% pumice plus nutrient, gypsum and Agri-lime). Control seedlings, after excising the root, were dipped in sterile water containing one drop of Tween-20® per liter. Pots were watered equally, and lids placed on top. Pots were transferred to a plant growth chamber (A1000, Conviron Asia Pacific Pty Ltd., Melbourne, Australia) set at 18 °C with a 16/8 h (light/dark) photoperiod. The experiment was laid out in a completely randomized design with eight replications. Each experimental unit comprised five pots/plants. After one month, plants were removed, and all soil debris cleaned by washing under a water tap. Each plant was placed on a filter paper for 8 h at room temperature (20–25 °C) to completely dry. The seedlings were weighed and the mean weight of five plants in each experimental unit were used for analysis of variance (ANOVA) using SPSS software (IBM© SPSS© Statistics, version 24).

## Dual culture test

The antifungal activity of a representative isolate of each bacterial species was assessed in vitro (dual culture) against the target pathogen, *L. maculans*. The *L. maculans* strain Lm145 utilized was a virulent pathogen of *Brassica* sp., previously isolated from a swede crop in New Zealand (*Lob, 2014*). Bacterial endophytes, previously stored at −80 °C, were thawed, streaked onto NA and incubated for 2 weeks at 22 °C. For each bacterial isolate a cell suspension ($10^9$ cells per mL) was prepared and 50 µL streaked in a straight line across the center of a Petri plate. Control treatments were streaked with only sterile water mixed with one drop of Tween-20® per liter. Petri plates were incubated for 2 weeks at 22 °C and then two mycelial plugs (5 mm diameter) were taken from the actively growing region of a 2-week old *L. maculans* culture and placed 25 mm from the edge of each Petri plate opposite each other; there were 10 replicate plates for each treatment. When the two fungal colonies placed on opposite sides had grown sufficiently to close the gap between them, the distance between the bacterial colony and the *L. maculans* colony was measured using a digital caliper (Mitutoyo Corporation, Kawaski, Kanagawa, Japan). The inhibition zone was rated using a 5 point scaling system: 4 (very high), inhibition zone >10 mm; 3 (high), imbibition zone 5–10 mm; 2 (medium), inhibition zone <5 mm; 1 (low), *L. maculans* growth stopped at the bacterial streak line; 0 (zero), no inhibition zone with *L. maculans* typically growing over the bacterial streak (*Hammoudi et al., 2012*).

# RESULTS

## Seed-originating bacterial isolates

In total, 54 accessions (44 wild and 10 landrace), out of the total 83 accessions surface disinfected and sown, resulted in symptomless plants when germinated and subsequently grown on MS medium. The remaining 29 accessions all exhibited epiphytic fungal growth, with most colonies identified as *Alternaria* sp. These plants were all destroyed by autoclaving. After incubation, bacterial colonies were detected from 48 symptomless wild *Brassica* accessions (38 wild and 10 landrace). Six accessions did not result in any bacteria being isolated.

## Identification of seed-originating bacteria

A sample of 90 bacterial isolates, representing the morphological diversity of all the observed bacterial colonies that developed from dissected *Brassica* tissue, were sequenced to determine their species identity. 16S rDNA gene sequencing identified 19 different bacterial species belonging to three phyla, namely Actinobacteria, Firmicutes and Proteobacteria (Fig. 1; Table 1). According to the phylogenetic tree (Fig. 1) the most frequently isolated species were *Methylobacterium fujisawaense* (50 isolates), *Stenotrophomonas rhizophila* (10 isolates) and *Pseudomonas lactis* (9 isolates). Three species were identified from the Actinobacteria, namely *Kocuria palustris*, *Micrococcus aloeverae*, and *Plantibacter flavus* while two species were identified from the Firmicutes phylum, namely *Bacillus mycoides* and *Paenibacillus hordei*.

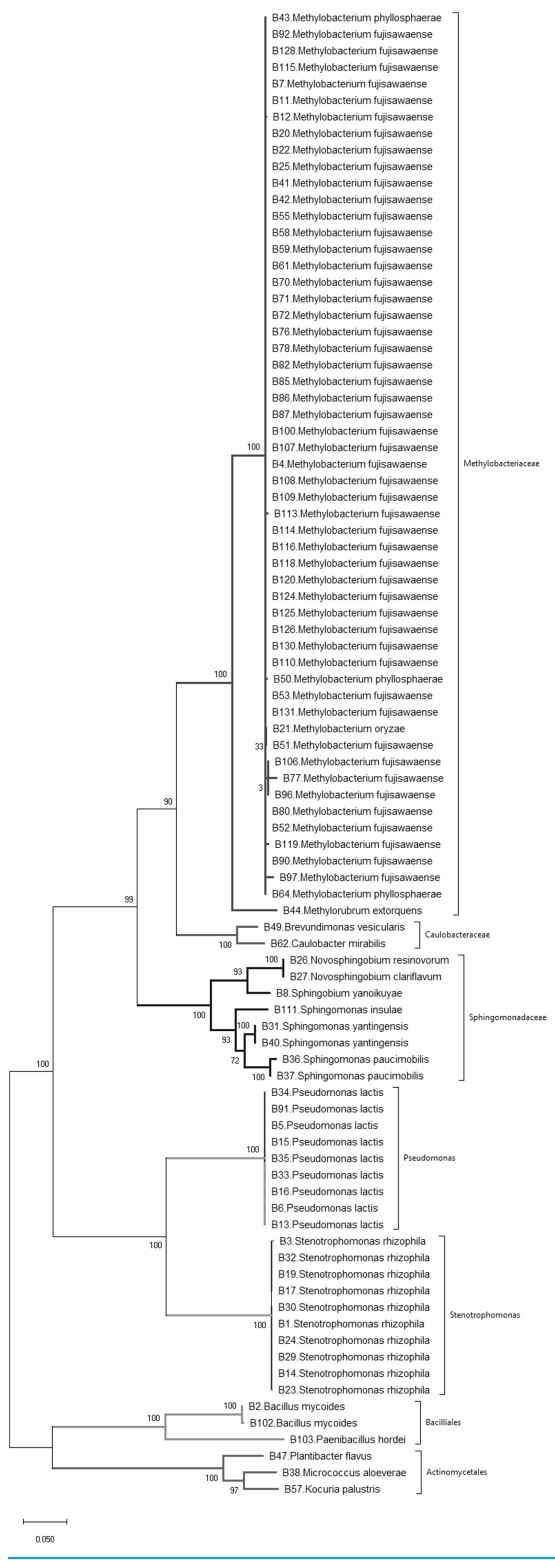

**Figure 1 Phylogenetic tree of bacteria isolated from wild and landrace *Brassica* accessions utilizing the maximum likelihood method, based on 16S rDNA gene sequences of isolates.** Bootstrap values were >55%.                

**Table 1  Seed-originating bacterial endophytes isolated from wild and landrace *Brassica* accessions.**

| Closest relative | Host species | Host origins |
|---|---|---|
| *Methylobacterium fujisawaense* | *Brassica barrelieri, Brassica elongate, Brassica juncea, Brassica nigra, Brassica rapa, Brassica napus, Brassica* sp. | Iceland, India, Iran, Italy, New Zealand, Portugal, Spain, Slovakia, Sweden, Sweden, USA, Zambia |
| *Methylobacterium oryzae* | *Brassica barrelieri* | Portugal |
| *Methylobacterium phyllosphaerae* | *Brassica incana, Brassica gravinae, Brassica napus* | Algeria, Finland, Iceland, New Zealand, Norway, Sweden, Ukraine |
| *Methylorubrum extorquens* | *Brassica rapa* | USA |
| *Stenotrophomonas rhizophila* | *Brassica juncea, Brassica oleracea* | Germany, Slovakia, Thailand |
| *Sphingomonas paucimobilis* | *Brassica balearica* | Unknown |
| *Sphingomonas yantingensis* | *Brassica juncea, Brassica* sp. | New Zealand, Slovakia |
| *Sphingomonas insulae* | *Brassica* sp. | New Zealand |
| *Pseudomonas lactis* | *Brassica balearica, Brassica juncea, Brassica nigra, Brassica oleracea* | Bulgaria, Germany, Thailand |
| *Sphingobium yanoikuyae* | *Brassica nigra* | Italy |
| *Bacillus mycoides* | *Brassica* sp., *Brassica juncea* | New Zealand, Slovakia |
| *Novosphingobium clariflavum* | *Brassica rapa* | Russia |
| *Novosphingobium resinovorum* | *Brassica rapa* | Russia |
| *Plantibacter flavus* | *Brassica juncea* | Mongolia |
| *Paenibacillus hordei* | *Brassica* sp. | New Zealand |
| *Kocuria palustris* | *Brassica napus* | Iceland |
| *Caulobacter mirabilis* | *Brassica napus* | Iceland |
| *Brevundimonas vesicularis* | *Brassica rapa* | USA |
| *Micrococcus aloeverae* | *Brassica* sp. | Unknown |

Simpson's diversity index recorded a value of 0.74 indicating that these wild and landrace *Brassica* accessions contained a high diversity of seed-originating bacteria. Nevertheless, *Methylobacterium* spp., predominantly *Methylobacterium fujisawaense* and the closely related species *Methylobacterium phyllosphaerae*, *Methylobacterium oryzae* and *Methylorubrum extorquens* constituted 56% of the bacteria isolated and were common in 77% of the *Brassica* accessions screened (38 accessions: 28 wild and 10 landrace). These were also distributed among multiple *Brassica* species including *Brassica barrelieri*, *Brassica elongate*, *Brassica gravinae*, *Brassica indica*, *Brassica juncea*, *Brassica napus*, *Brassica nigra* and *Brassica rapa* sourced from five continents, namely Africa, Asia, Australasia, Europe and North America.

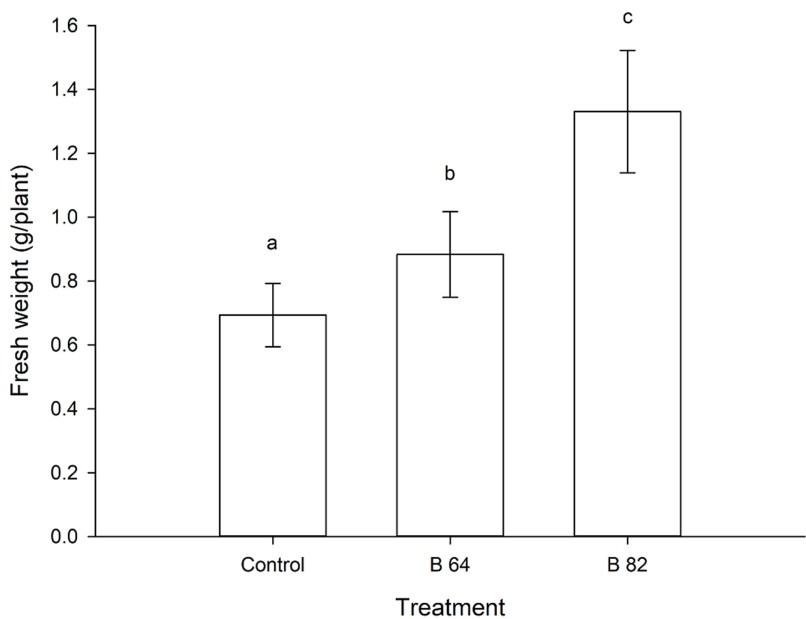

**Figure 2 Mean fresh weight of oilseed rape plants inoculated with *Methylobacterium phyllosphaerae* (B64) and *Methylobacterium fujisawaense* (B82) plus uninoculated (control) plants (± SD).** Bars followed by the same letter are not significantly (*P* > 0.05) different according to Fisher's protected LSD test.                                    

## Plant growth promotion

Oilseed rape plants that were inoculated with two isolates of *Methylobacterium* showed a significant (*P* < 0.01) increase in their fresh weight when compared to the uninoculated control plants (Fig. 2). Plants inoculated with *Methylobacterium fujisawaense* (B82) had a mean weight of 1.33 g/plant, while those inoculated with *Methylobacterium phyllosphaerae* (B64) displayed a mean weight of 0.88 g/plant compared to a mean weight of 0.69 g/plant for the uninoculated control plants (Fig. 2).

## Dual culture test

The results from the dual culture assays indicate that some of the bacterial isolates showed antagonistic behavior towards *L. maculans*. The strongest inhibition effect was observed with isolates of *Methylobacterium fujisawaense* and *Methylobacterium phyllosphaerae*, *N. resinovorum*, *Pseudomonas lactis*, *Plantibacter flavus* and *Stenotrophomonas rhizophila* that all showed a high, clear inhibition zone between the edge of the bacterial streak and the pathogen (Fig. S1). However, *L. maculans* was not affected by *K. palustris*, *Sphingomonas yantingensis*, *Sphingomonas insulae*, *Bacillus mycoides* or *Brevundimonas vesicularis*; with no inhibition zone produced.

## DISCUSSION

This study investigated the cultivable bacterial community persisting in wild and landrace *Brassica* seed. Nineteen bacterial species were isolated from 83 accessions, belonging to eight *Brassica* species, covering five continents, with some of the accessions more than 20 years old. The bacterial genera to which these species belong have been previously

reported in the literature as seed endophytes of a diverse number of plant species. For example, *Methylobacterium* and *Paenibacillus* spp. have both been described as seed endophytes from *Eucalyptus* (*Ferreira et al., 2008*), *Oryza sativa* (*Mano et al., 2006*) and *Phaseolus vulgaris* (*López-López et al., 2010*), while *Bacillus* and *Micrococcus* spp. are common seed endophytes of *Coffea arabica* (*Vega et al., 2005*) and *O. sativa* (*Mano et al., 2006*). Our results indicate that the diversity of bacterial endophytes in seed of wild *Brassica* is relatively high with most of the bacterial species identified belonging to the Proteobacteria, the major phylum of gram-negative bacteria. This is consistent with earlier work that showed the seed microbiome of oilseed rape were colonized mostly by Proteobacteria and that individual cultivars each had their own unique microbiome profile (*Rybakova et al., 2017*).

Development of an effective surface disinfection protocol was paramount to this study. A protocol that was too harsh could sterilize the seed and kill any potentially beneficial microorganisms residing in the seed tissues, as well as potentially damaging the seed itself, while a protocol that was too moderate could yield unwanted saprophytic microorganisms residing on the surface of the seed coat. These non-target saprophytes have the potential to outgrow any slower growing endophytic organisms that may be beneficial. Many of these non-target species can colonize the interior tissues of the germinating plant during the emergence of the radicle (*Bent & Chanway, 2002*). The surface disinfection protocol used in this study was not designed to eliminate all organisms living on the seed surface, just to reduce their frequency. For example, *Alternaria* sp., commonly associated with seed coats or pericarps of seed (*Harman, 1983*; *Neergaard, 2011*) was frequently isolated.

Many studies have reported that strains belonging to the same genera identified in our study confer several beneficial traits to their host plants, including enhanced resistance against certain plant pathogens and/or growth promotion (*Araújo et al., 2002*; *Berg & Hallmann, 2006*; *Khan et al., 2014*; *Rashid, Charles & Glick, 2012*; *Rout & Chrzanowski, 2009*; *Sessitsch, Reiter & Berg, 2004*; *Ying et al., 2016*). We assessed the antagonistic activity of selected isolates of bacterial species against *L. maculans* (the causal agent of phoma stem canker in oilseed rape) through dual culture bioassays and observed that *Methylobacterium fujisawaense* and *Methylobacterium phyllosphaerae* possessed antagonistic potential against the pathogen. The genus *Methylobacterium* is composed of pink-pigmented facultative methylotrophs (PPFMs) (*Dourado et al., 2015*) that are able to form endophytic associations with a range of plant species including citrus (*Araújo et al., 2002*), cotton (*Madhaiyan et al., 2012*), eucalyptus (*Andreote et al., 2009*), mangrove (*Dourado et al., 2012*), peanut (*Madhaiyan et al., 2006b*), pine (*Pohjanen et al., 2014*), tobacco (*Andreote et al., 2006*) and white cabbage (*Wassermann et al., 2017*). PPFMs are not pathogenic to their plant hosts (*Idris et al., 2006*) making them ideal candidates for endophytic biological control strategies (*Omer, Tombolini & Gerhardson, 2004*). Additionally, *Methylobacterium* spp. are able to enhance plant growth through several mechanisms, including, nitrogen fixation (*Lee et al., 2006*; *Menna et al., 2006*; *Sy et al., 2001*), phytohormone production such as cytokinins and auxins (*Madhaiyan et al., 2006a*; *Meena et al., 2012*; *Trotsenko, Ivanova & Doronina, 2001*), interact with and

inhibit plant pathogens (*Araújo et al., 2002*; *Lacava et al., 2004*; *Poorniammal, Sundaram & Kumutha, 2009*), promote plant growth (*Madhaiyan et al., 2006a*, *2006b*; *Tani et al., 2012*), induce higher photosynthetic activity (*Cervantesmartinez, Lopezdiaz & Rodriguezgaray, 2004*), induce systemic resistance (*Madhaiyan et al., 2006b*), decrease environmental stress (*Muller et al., 2011*) and immobilize heavy metals (*Dourado et al., 2012*). We analyzed the fresh weight of seedlings of an oilseed rape cultivar under growth chamber conditions when the roots were inoculated with two isolates of *Methylobacterium fujisawaense* and *Methylobacterium phyllosphaerae* and found that inoculated plants had a higher growth rate than non-inoculated plants. Cultivated *Brassica* crops, such as oilseed rape, have a high nitrogen demand (*Rathke, Behrens & Diepenbrock, 2006*) and their cultivation is reliant on fertilization with nitrogen rich products. These crops usually have low nitrogen use efficiency and this is a specific target for the breeding of new cultivars (*Bouchet et al., 2014*, *2016*). The frequent presence of *Methylobacterium* in wild *Brassica* species, that are usually found within infertile soils, such as those where some of the wild species used in our study were collected suggests that this symbiosis improves the development of the host plant. These bacteria may therefore possess traits for use as plant growth promoters in artificial *Brassica* hosts such as domesticated cultivars.

This study isolated species of *Methylobacterium*, and the closely related *Methylorubrum extorquens* (*Green & Ardley, 2018*), from above and below ground plant organs (shoot and root, respectively). As morphologically similar isolates were identified from multiple root and shoot tissue pieces belonging to the same individual plant, we speculate that these bacterial isolates are capable of systemic plant colonization. Additionally, these tissues were dissected from symptomless seedlings grown from surface disinfected seed under sterile conditions and therefore this strongly suggests that these bacteria are vertically transmitted. The *Methylobacterium* species isolated in this study were present in a range of plant accessions originating from a geographically diverse set of countries with varied altitude. This is consistent with other reports of endophytic microbes, for example, among *Zea* spp. which were found across species grown in wide range of geographical locations (*Johnston-Monje & Raizada, 2011*).

It has been reported the age of seed may considerably influence the seed microbiome (*Cankar et al., 2005*). Indeed, no bacteria were isolated from six accessions that had been stored for more than 15 years. However, one accession that was over 26 years old gave rise to *Methylobacterium* indicating that this bacterium can adapt and survive in seed tissues for a long period of storage time. *Mano et al. (2006)* reported that only certain bacteria such as *Methylobacterium* are able to reside inside rice seed. These endophytic bacteria enter the seeds during the seed maturation stages and are tolerant to high osmotic pressure. The isolates possess a high degree of amylase activity, which may aid survival in the seed (*Mano et al., 2006*).

## CONCLUSIONS

Although three species of *Methylobacterium*, namely *Methylobacterium extorquens*, *Methylobacterium mesophilicum* and *Methylobacterium goesingense*, were previously

identified in *Thlaspi goesingense* belonging to the wider *Brassicaceae* family (*Idris et al., 2006*), to our knowledge this is the first report to describe the isolation and identification of endophytic bacteria of seeds of wild and landrace *Brassica* species. We present a straight-forward strategy to screen and cultivate seed-originating endophytes with possible beneficial traits.

The microbiome of many vegetables, including *Brassica* spp., can serve as sources of biological control agents (*Wassermann et al., 2017*) while focusing our efforts on seed-originating organisms may facilitate novel endophyte technologies that could be incorporated into future crop seed (*Berg et al., 2017*). This approach would then also be advantageous to companies that wish to invest in the commercialization of such products as they can lower their financial risk in terms of delivering a suitable efficacious product to farmers whilst protecting their IP. The latter is possible because an elite plant cultivar and the biological control agent can be protected together in one commercial seed product entity. This means of propagation relies on the plant's reproductive strategy and may aid the marketing of any potential plant-endophyte product (*Card et al., 2016*).

## ACKNOWLEDGEMENTS

We would like to express our appreciation to Jaspreet Singh and Anouck de Bonth (both from AgResearch Limited) and Jana Monk (AsureQuality Limited, New Zealand) for their technical support. We thank Eirian Jones (Lincoln University, New Zealand) for kindly supplying the culture of *Leptosphaeria maculans* and DSV seeds for providing oilseed rape, cv. King.

### Funding

This project was supported by Grasslanz Technology Limited (New Zealand), the TR Ellet Agricultural Research Trust, the George Mason Sustainable Land Use scholarship and The Agricultural Research, Education and Extension Organization (AREEO), Iran.
The funders had no role in study design, data collection and analysis, decision to publish, or preparation of the manuscript.

### Grant Disclosures

The following grant information was disclosed by the authors:
Grasslanz Technology Limited, New Zealand.
TR Ellet Agricultural Research Trust.
George Mason Sustainable Land Use scholarship.
The Agricultural Research, Education and Extension Organization (AREEO), Iran.

### Competing Interests

Davood Roodi is employed by Seed and Plant Improvement Institute. Stuart D. Card, Richard D. Johnson and Ruy Jauregui are employed by AgResearch Limited.

## Author Contributions

- Davood Roodi conceived and designed the experiments, performed the experiments, analyzed the data, prepared figures and/or tables, and approved the final draft.
- James P. Millner analyzed the data, authored or reviewed drafts of the paper, and approved the final draft.
- Craig McGill analyzed the data, authored or reviewed drafts of the paper, and approved the final draft.
- Richard D. Johnson conceived and designed the experiments, analyzed the data, authored or reviewed drafts of the paper, and approved the final draft.
- Ruy Jauregui analyzed the data, prepared figures and/or tables, and approved the final draft.
- Stuart D. Card conceived and designed the experiments, performed the experiments, analyzed the data, prepared figures and/or tables, authored or reviewed drafts of the paper, and approved the final draft.

## DNA Deposition

The following information was supplied regarding the deposition of DNA sequences:

All sequence data are available in GenBank: MN629046–MN629135.

## Data Availability

The raw data containing information related to all seed accessions, such as original host genebank, country of origin and age, is available in the Supplemental Files.

## Supplemental Information

Supplemental information for this article can be found online at http://dx.doi.org/10.7717/peerj.9514#supplemental-information.

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
