# Peer review of "Methylobacterium, a major component of the culturable bacterial endophyte community of wild Brassica seed"

_PeerJ, doi:10.7717/peerj.9514_

## Round 0.1 · original submission · Minor Revisions

Please address all the reviewers concerns, specially explain in the text to reviewer 2, why culturable collections still very valuable for science.

·

Basic reporting

The article was written in clear English. However, the paper has some problems in punctuation marks and font style. For example, lines 38 and 51 of the abstract.

There are some misspellings (lines 188, 192, 197, 199, 202, 212and 214).

In the results of the “Plant growth promotion” section (line: 254), you report the increase of the growth in dry weight. However, on the axis of the figure 2 was noted like “fresh weight”. Also, check the raw data table.

References: you must correct the molecular nitrogen annotation (lines 484 and 562).

All these observations were noted in the original text. This document will be attached in PDF format.

Experimental design

In materials and methods, in the section “Identification of seed-associated bacterial endophytes” (line: 156), the authors used partial 16S rDNA sequences in agreement to Weisburg et al. (1991). However, it is important that you analyze the work of Kato et al. 2005 (J. Gen. Appl. Microbiol., 51, 287–299). Because, the HV-region of 16S rDNA used in this study produced a better grouping of Methylobacterium species. In addition, this region has been used in other genres and produces the same results (for example: Bacillus, Goto et al. 2002). Maybe you could adjust the length of their sequences and produce a new tree.

Validity of the findings

You should add images of the “Dual culture test” (line: 261). I believe that these results are important to confirm the importance of Methylobacterium like antifungal.

Reviewer 2 ·

Basic reporting

Endophytes are important plant inhabitants that f.e. provide tolerance against biotic and abiotic stress. They were discovered and defined in the 80th; methods of microbiome research (multi-omics, advanced microscopy) boost our knowledge about endophytes and the seed microbiome during the last ten years. However, this study based on a traditional cultivation approach of endophytes of Brassica seeds.Such approach is of value for applied research, e.g. looking for novel plant growth promoting or biocontrol bacteria. For basic research, methods are not state of the art.

Experimental design

Authors studied seeds of 83 wild and landrace Brassica accessions composed of 14 different species with a worldwide distribution for seed-associated bacterial endophytes.
However, I see problems with experimental design in the following points:
• The methods applied are very traditional, and based on cultivation of microbes only. A combined approach e.g. with 16S rDNA amplicon sequencing would be of value and allow a much better assessment of the cultivated strains.
• The design of the seed samples is not well explained. There is an impact of seed age and origin, which is not considered.
• 29 accessions were excluded due to exhibited epiphytic fungal growth of Alternaria sp.

Validity of the findings

Due to the problems with the experimental design I don’t think that the conclusions are really representative.

Other comments:
Abstract: Microbial species cannot be parasitic. Half of the abstract is introduction only, on the other side half of the experiments is not yet mentioned in the abstract (PGPR by Methylobacteria).
"Seed-associated" is not correct because the authors studied the phyllosphere and rhizosphere of plants - "seed-originating" would be better. Despite the differentiation of isolates according to their habitat, this is not well considered in the results and discussion section but would be interesting.
Abbreviations of the genera are not correct.
Missing important references:
Hardoim PR, van Overbeek LS, Berg G, Pirttilä AM, Compant S, Campisano A, Döring M, Sessitsch A. 2015. The Hidden World within Plants: Ecological and Evolutionary Considerations for Defining Functioning of Microbial Endophytes. Microbiol Mol Biol Rev. 79(3):293-320
Wassermann B, Rybakova D, Müller C, Berg G. 2017. Harnessing the microbiomes of Brassica vegetables for health issues. Sci Rep. 7(1):17649.

Additional comments

Best wishes Gabriele

---

## Round 0.2 · accepted · Accept

I consider that all the issues raised by reviewers where correctly address and the ones where authors did not agree were properly justified, this is a nice study and is ready for publication.